# Cross-Reactivity between Chemical Antibodies Formed to Serum Proteins and Thyroid Axis Target Sites

**DOI:** 10.3390/ijms21197324

**Published:** 2020-10-03

**Authors:** Datis Kharrazian, Martha Herbert, Aristo Vojdani

**Affiliations:** 1Department of Neurology, Harvard Medical School, Boston, MA 02115, USA; martha.herbert@mgh.harvard.edu; 2Massachusetts General Hospital, Charlestown, MA 02129, USA; 3Department of Preventive Medicine, Loma Linda University School of Medicine, Loma Linda, CA 92354, USA; drari@msn.com; 4Immunosciences Laboratory, Inc., Los Angeles, CA 90035, USA

**Keywords:** hapten, neoantigen, cross-reactivity, thyroid, chemicals

## Abstract

In some instances, when chemicals bind to proteins, they have the potential to induce a conformational change in the macromolecule that may misfold in such a way that makes it similar to the various target sites or act as a neoantigen without conformational change. Cross-reactivity then can occur if epitopes of the protein share surface topology to similar binding sites. Alteration of peptides that share topological equivalence with alternating side chains can lead to the formation of binding surfaces that may mimic the antigenic structure of a variant peptide or protein. We investigated how antibodies made against thyroid target sites may bind to various chemical–albumin compounds where binding of the chemical has induced human serum albumin (HSA) misfolding. We found that specific monoclonal or polyclonal antibodies developed against thyroid-stimulating hormone (TSH) receptor, 5′-deiodinase, thyroid peroxidase, thyroglobulin, thyroxine-binding globulin (TBG), thyroxine (T4), and triiodothyronine (T3) bound to various chemical HSA compounds. Our study identified a new mechanism through which chemicals bound to circulating serum proteins lead to structural protein misfolding that creates neoantigens, resulting in the development of antibodies that bind to key target proteins of the thyroid axis through protein misfolding. For demonstration of specificity of thyroid antibody binding to various haptenic chemicals bound to HSA, both serial dilution and inhibition studies were performed and proportioned to the dilution. A significant decline in these reactions was observed. This laboratory analysis of immune reactivity between thyroid target sites and chemicals bound to HSA antibodies identifies a new mechanism by which chemicals can disrupt thyroid function.

## 1. Introduction

Immunological cross-reactivity is expressed when antibodies with similar amino acid homology or similar antibody surface topology bind to the same binding site [1,2]. The interactions of multiple antigenic antibodies with the same binding site are known as cross-reactivity [3]. Cross-reactivity of various antigens with self-tissue proteins can induce tissue-specific autoimmune diseases in susceptible subjects [1,2]. These molecular interactions with the antigen–antibody binding sites can occur from a diverse list of antigen-promoted antibodies [4]. Cross-reactive antibodies from various infections have been found to play a role in autoimmune thyroid disease and thyroid metabolism dysfunction by binding to multiple target sites of the thyroid axis via cross-reactivity [5,6,7,8,9,10,11,12,13,14]. Furthermore, many antibody binding sites are polyfunctional and can accommodate more than one antigenic epitope and play a role in autoimmune disease [15]. Cross-reactive interactions with various target sites of the thyroid axis may also lead to thyroid metabolism disruption.

Reactivity of antibodies with chemicals bound to proteins has the potential to play a cross-reactive role in autoimmune thyroid disease and thyroid metabolism disruption. In addition, the binding of chemicals to self-proteins such as albumin, globulin, or hemoglobin leads to protein misfolding and induces a conformational change in the macromolecule. The alteration of protein topography leads to the binding of the antibody to the protein in the target sites [16,17]. Chemical molecules can bind directly or indirectly to circulating proteins after hepatic or extrahepatic conversion from prohapten to haptens, generating hapten–protein adducts. This leads to neoantigen formation, resulting in systemic T-cell or antibody immune responses against the haptens and self-proteins [16,18].

Conjugation of chemicals bound to human serum albumin (HSA) is found with blood samples of healthy human blood donors. In a study we published of 400 subjects, between 8–22% of individuals exhibited elevated levels of commonly exposed chemicals bound to human serum albumin [19]. Further research identified that these chemicals conjugated to HSA are associated with antibodies to neurological tissue involved in multiple sclerosis [20]. We also determined that elevated levels of bisphenol-A bound to HSA significantly increases the risk for Parkinson’s disease and correlates with alpha-synuclein antibodies [21]. Studies on thyroid cross-reactivity have focused primarily on pathogens and dietary proteins [22,23]. The study of chemical cross-reactivity and thyroid disruption has not been thoroughly investigated.

In this laboratory study, we evaluated the potential for anti-thyroid axis antibodies to bind to various chemical–albumin complexes that have chemically-induced human serum albumin (HSA) misfolding; this misfolding leads such compounds to act immunologically similarly to thyroid target site antigens such as thyroid-stimulating hormone (TSH) receptor, 5′-deiodinase, thyroid peroxidase, thyroglobulin, thyroxine-binding globulin (TBG), thyroxine (T4), triiodothyronine (T3), and various chemicals bound to albumin. 

Cross-reactivity between chemically-induced misfolded HSA with the various target sites of the thyroid axis may play a role in the pathophysiology of thyroid autoimmunity. This could impact various aspects of thyroid metabolism, which could interfere with the proper dosage of thyroid hormones, impair thyroid peripheral metabolism, disrupt thyroid feedback loops, and alter thyroid hormone transport. We evaluated the immune reactivity of chemicals bound to human albumin, including chemical compounds found in plastic products, foams, cosmetics, upholstery, dry cleaning agents, fire retardants, metal products, and chemicals commonly found in water and soil contamination. In this study, we investigate whether chemicals bound to human serum albumin can develop into neoantigens that cross-react with thyroid axis sites.

## 2. Results

### 2.1. Results of ELISA

Eleven chemicals bound to HSA and eleven chemicals in native form (unconjugated) were tested with seven target tissue antibodies leading to 84 antigen–antibody OD measurements using ELISA methods. The enzyme protein disulfide isomerase (PDI) was also tested with seven target tissue antibodies suing ELISA. The OD range for all of the chemicals bound to HSA was 0.1–2.02. The mean OD was 0.46, with a standard deviation of 0.45. The OD of 0.91 represented one standard deviation from the mean of all chemicals measured that were bound to HSA. The OD of 1.35 represented two standard deviations from the mean of all chemicals measured that were bound to HSA. The OD of 1.81 represented three standard deviations from the mean of all chemicals measured that were bound to HSA. Additionally, control studies were conducted for each of the seven tissue antibodies with OD measurements that were all found to be less than 0.2. Significant OD measurements were defined as those measurements that were above the control OD and at least 1 standard deviation from the mean of all single measurements of the 84 antigen–antibody ODs. These findings are illustrated as red bars in Figure 1, Figure 2, Figure 3 and Figure 4.

Immunological reactivity using monoclonal and polyclonal antibodies made against thyroid target antigens showed various degrees of reaction (Figure 1, Figure 2, Figure 3 and Figure 4). We identified immune reactivity with thyroid-stimulating hormone receptor (TSH-R). The control wells coated with HSA, BSA, or hemoglobulin were less than 0.2 OD. The OD for formaldehyde-HSA (F-HSA) was 0.47. Aflatoxin bound to albumin OD was 0.9 (1+), and isocyanate bound to albumin OD was 1.3 (2+). The reactions with TSH-R for nine other chemicals were less than 0.2, which is equivalent to the control wells (Figure 1).

Immune reactivity was identified with thyroxine deiodinase antibody with all tested chemicals, with ODs ranging from 0.4 OD for T-BPA-HSA to 0.7 OD for TPA-HSA in comparison to control. The OD for aflatoxin bound to albumin was 1.1 (1+) and formaldehyde bound to albumin was 0.95 (1+), which were more significant (Figure 2).

Immune reactivity with T3 for all tested chemicals was above the control OD (0.2) and ranged from 0.78 for TPA-HSA to 2.0 for parabens-HSA. Specifically, the OD for aflatoxin bound to albumin was 1.6 (2+), formaldehyde bound to albumin OD was 1.3 (1+), isocyanate bound to albumin OD was 1.0 (1+), 2,4-dinitrophenol bound to albumin OD was 0.7 (1+), protein disulfide isomerase OD was 2.1 (3+), bisphenol-A OD was 1.3 (1+), TBBTA bound to albumin OD was 1.3 (2+), tetrachloroethylene bound to albumin OD was 1.5 (1+), mercury bound to albumin OD was 1.0 (1+), parabens bound to albumin OD was 2.0 (3+), and heavy metal composite bound to albumin (2+) OD was 1.7 (Figure 3).

Immunological reactivity with thyroglobulin antibody only identified a reaction with aflatoxin bound to albumin with a significant OD of 1.5 (2+). The other 11 chemicals did not demonstrate a significant reaction compared to the control (Figure 4).

Immunological reactivity between thyroid peroxidase (TPO), TBG, and T4 antibodies resulted in ODs that were not significant and below 0.5. There was a borderline reaction between TPO antibody and formaldehyde-HSA with an OD of 0.46. The ODs for TPO for the other 11 reactions were much lower and comparable to the control OD.

### 2.2. Demonstration of Analytical Specificity of Thyroid Antibodies Binding to Different Haptenic Chemicals to HSA

The analytical specificity of these polyclonal and monoclonal antibodies binding to haptenic chemicals bound to HSA was confirmed by serial dilation and inhibition studies.

Figure 5 and Figure 6 show the binding of different serially diluted antibodies made against thyroid antibodies to different haptenic chemicals bound to HSA. In proportion to the dilutions, there was a significant decline in the ODs that were observed when the reaction of thyroid antibody to haptenic chemical bound to HSA was high. For example, thyrotropin receptor (TSH-R) antibody binding to isocyanate-HSA at the dilution of 1:100 resulted in OD of 2.5, 1:200 resulted in OD of 1.5, 1:800 resulted in OD of 0.6, and 1:1200 resulted in OD less than 0.2 (Figure 5). Thyroglobulin antibodies to aflatoxin-HSA at a dilution of 1:100 resulted in OD of 2.6, 1:200 resulted in OD of 2.1, 1:800 resulted in OD of 0.8, and 1:1200 resulted in OD of less than 0.3 (Figure 6).

Inhibition studies conducted with all native chemicals as well as with HSA alone demonstrated no ability to inhibit any of the thyroid antibodies. Inhibition of thyroid antibodies was only found with haptens complexed with HSA. Figure 7, Figure 8, Figure 9 and Figure 10 demonstrate inhibition of thyroid antibodies binding to haptenic chemical-coated plates by different concentrations of the haptenic chemicals. For example, for TSH-R antibody to isocyanate-HSA antibody, the addition of 2 μg/mL of isocyanate-HSA resulted in 0% inhibition of anti-TSH-R antibody to isocyanate-HSA-coated plates. However, the addition of 4, 16, or 64 μg/mL of isocyanate-HSA to the liquid phase of ELISA resulted in 15, 46, or 76% inhibition of this antibody–antigen reaction, respectively (Figure 7). Similar inhibition of binding was observed when aflatoxin was added to the reaction, but not with BPA bound to HSA or with 2,4-Din-HSA for TSH-R antibodies. These findings further support the specificity of aflatoxin-HSA and isocyanate-HSA binding to TSH-R, as shown in Figure 1.

Furthermore, comparing the inhibition of antibody binding of thyroxine deiodinase after the addition of a high concentration of aflatoxin bound to HSA and formaldehyde bound to HSA resulted in significant inhibition of these antibody–antigen reactions. The addition of parabens bound to HSA in the liquid phase caused minimum inhibition of the T4 antibodies to the paraben-bound-to-HSA-coated plates as shown in Figure 8. These findings further support the specificity of aflatoxin-HSA and formaldehyde-HSA to thyroxine deiodinase, as shown in Figure 2.

In Figure 9 and Figure 10, similar results were obtained only when haptenic chemicals bound to HSA reacted strongly with a particular thyroid antibody; when the same chemical was added to the liquid phase of the ELISA, it resulted in significant inhibition. Furthermore, when the reaction of thyroid antibody resulted in low ODs with specific haptenic chemicals bound to HSA, even high concentrations of the same chemical added to the liquid phase did not cause inhibition in the antibody–antigen reaction (Figure 11).

## 3. Discussion

Our study’s goal was to investigate whether chemicals bound to human serum albumin can develop into neoantigens that cross-react with thyroid axis sites. Our study found that antibodies made against TSH-R, DIO2, thyroglobulin, and T3 interact with chemicals bound to HSA (Figure 1, Figure 2, Figure 3 and Figure 4). Previous models of how chemicals play a role in disrupting thyroid metabolism include binding of the chemicals with nuclear hormone receptors, orphan and neurotransmitter receptors, and direct chemical alteration of enzymatic pathways [24]. Our study identified a new mechanism through which chemicals bound to albumin lead to structural protein misfolding, which creates neoantigens that lead to the development of antibodies that bind to key target proteins of the thyroid axis through cross-reactivity. The outcomes of this study may serve to fill a knowledge gap related to the way chemicals may influence thyroid function through a novel pathway that has not yet been identified in the thyroid literature to the best of our knowledge.

As chemicals bind to proteins and induce a conformational change in the macromolecule, there is the potential to create new structures that may react with various target sites of antibodies [16]. Cross-reactivity can occur if epitopes of the protein share surface topology to similar binding sites [17]. Alterations of peptides that share topological equivalence of alternating side chains can lead to the formation of binding surfaces that can mimic the antigenic structure of a variant peptide [25]. In addition to peptide similarity with models of cross-reactivity, it is now clear that binding sites are polyfunctional and can accommodate more than one antigenic epitope [26]. The cross-reactive binding of specific epitopes with thyroid axis target sites may explain a previously uninvestigated mechanism of the way chemicals may interfere with thyroid hormone metabolism, interfere with thyroid medication dosage, or potentially promote autoimmune reactivity.

In our study, several chemicals bound to albumin exhibited specific cross-reactivity with T3. These chemicals included aflatoxins, formaldehyde, isocyanate, 2,4-dinitrophenol, protein disulfide isomerase, BPA, and TBBPA. The structural similarity between them may explain their roles as both thyromimetics and immunological cross-reactive target proteins. The structure of thyromimetics is based on those of endogenous thyroid hormones, which consist of a biaryl ether skeleton substituted with iodine, alpha-alanine moiety, and a hydroxyl group at two 2,4-dinitrophenol rings [27].

Interestingly, our study found that antibodies made against T3 reacted with several chemicals, but antibodies made against T4 did not. Although T3 and T4 have some structural similarity, there are significant biochemical structural differences between them. When differentiated from T3, there are two conformations that are independent with T4 in the outer phenyl ring structure and independent conformations in the crystal lattice. The significant distinctions between T3 and T4 structures are condensed C4’-O4’ bond contraction of the C3’-C4’-C5’ angle and an enlargement in the C3’ and carbon C5’ angles of T4 [24]. These differences may explain why antibodies made against T3 but not against T4 reacted with chemicals bound to HSA. Due to the fact that T3 has structural similarity to thyromimetics, such as BPA and TBBPA, it is possible that the cross-reactivity with T3 antibodies occurred unrelated to protein misfolding of these secondary epitopes with HSA. However, our specificity studies found no inhibition of T3 antibodies with native adjunct alone. Therefore, we propose that despite T3 structural similarity with various chemicals, the production of secondary epitopes was responsible for our findings of cross-reactivity. This chemical–protein adduct cross-reactivity with T3 may impact thyroid medication dosage and impact thyroid metabolism by interfering with T3 directly in the periphery.

Our investigation found immune reactivity between BPA and antibody to T3. This may suggest that immune reactivity to compounds in plastic products impacts circulating T3 levels and thyroid autoimmunity. BPA and T3 possess such a high degree of molecular structure similarity that BPA may act as an antagonist compound on T3 receptor sites [28]. In particular, hydrocarbon rings found both on BPA and T3 with anchor ring-like similarities may induce immune reactivity [29,30,31,32]. When compounds have structural similarity, it may potentially lead to immune reactivity with the formation of antigen–antibody complexes, which then may result in inflammation and autoimmunity [33]. The potential for BPA to induce thyroid autoimmunity due to immune reactivity has been previously reported [34]. In our study, we found that antibodies formed against BPA bound to HSA may play a role in cross-reactivity with T3, leading to potential thyroid metabolism disruption.

Additionally, we identified cross-reactivity between anti-T3 antibody and the enzyme PDI. PDI directly acts to catalyze protein folding and the multimerization of thyroglobulin in the follicular lumen of the thyroid gland [35,36]. PDI also has a role in the biosynthesis of T3 by inactivating type 2 iodothyronine 5′-deiodinase involved with converting T4 into bioactive T3 [36,37,38,39]. BPA binds to PDI, which is located throughout the body and potentially accountable for the diverse list of physiological influences of BPA due to enzyme function disruption [39]. Therefore, demonstration of cross-reactivity between T3 and PDI may explain how the binding of BPA to PDI could be another mechanism of interference with thyroid function, due to cross-reactivity, as we identified in our study.

TBBPA is a fire-retardant compound. Our study found that when TBBPA binds to HSA it forms a conformational change in the peptide that exhibits cross-reactivity reactions with T3. It should be noted that TBBPA has already been found to share structural similarity to thyroid hormones and can interfere with thyroid hormone physiology [40]. This structural similarity between TBBPA and T3 not only allows it to compete with thyroid receptor sites, but also allows for potential immunological cross-reactivity, as identified in our study.

We identified cross-reactivity between T3 and 2,4-dinitrophenol bound to HSA. Research shows 2,4-dinitrophenol has structural similarity to thyroid hormones and hence has an affinity to bind to thyroid transport proteins [41]. Exposure to 2,4-dinitrophenol has been reported to induce thyroid insufficiency. The immunological mechanisms of cross-reactivity that we identified in our study between 2,4-dinitrophenol bound to HSA and T3 may explain the mechanism behind thyroid metabolism disruption that is reported in the literature [42,43].

Our study also identified cross-reactivity between TSH-R and T3 with isocyanates-HSA. Several studies have demonstrated that isocyanates have the potential to disrupt thyroid metabolism [44,45,46]. It is possible that the structural similarity of isocyanates to T3 leads to its acting as an endocrine disruptor and to its playing a potential cross-reactive role, which is identified in our study with TSH-R and T3. A similar mechanism of structural similarity between formaldehyde and T3 may occur, as our study identified cross-reactivity between formaldehyde-HSA and both T3 and thyroxine 5-deiodinase. These immunological reactions may explain the findings of an animal study in which formaldehyde exposure altered thyroid function and reduced circulating T3 levels [47].

Immunological reactivity was identified between formaldehyde-HSA with thyroxine 5-deiodinase and T3. In a small animal study, rats exposed to formaldehyde showed alterations in thyroid function and reduced T3 levels [43]. The reaction between antibodies to T3, thyroxine 5-deiodinase, and formaldehyde bound to HSA suggests this immunological reactivity has the potential to interfere with circulating T3 levels and thyroid conversion of T4 and T3.

The specificity of these thyroid antibodies to haptenic chemicals binding to HSA was shown by serial dilution of the antibodies added to the chemicals bound to HSA in the solid phase, and by the addition of different concentrations of haptenic chemicals in the liquid phase and examining decline in the ODs that are proportioned to the concentrations. A key question to ask in our study is whether the chemical adduct itself or the modifying effect of the chemical adduct bound to human serum albumin had the ability the inhibit the binding of antibodies. In our laboratory study, we found no inhibition of antibody binding to the chemicals alone. Inhibition only occurred with the adduct formed by the HSA complexed with chemicals. Furthermore, we found that even high concentrations of chemicals bound to HSA did not inhibit the reactions when the optical density was at least in the range of 0.7 (Figure 11). These findings shows that there is some degree of non-specific reaction between thyroid antibodies and chemicals bound to HSA; otherwise, the addition of chemicals in the liquid phase should have inhibited these antigen–antibody reactions. These findings provide support that it is the complexed chemicals and HSA that is specifically reacting to thyroid antibodies and not native chemicals. Dose-responsive curves of these studies are shown in Figure 7, Figure 8, Figure 9, Figure 10 and Figure 11, and these findings support the specificity of our thyroid binding of haptenic chemicals to HSA as shown in Figure 1, Figure 2, Figure 3 and Figure 4.

## 4. Materials and Methods

### 4.1. Polyclonal and Monoclonal Antibodies

We purchased mouse monoclonal antibody to TBG with purified human TBG as the immunogen, mouse monoclonal antibody thyroglobulin that was raised in mouse using human thyroglobulin as the immunogen, mouse monoclonal antibody thyroxine that was raised in mouse using thyroxine (T4)–bovine serum albumin (BSA) as the immunogen, mouse monoclonal triiodothyronine antibody that was raised in mouse using triiodothyronine BSA as the immunogen, and mouse monoclonal thyroid peroxidase antibody that was raised in mouse using human thyroid peroxidase as the immunogen from MyBioSource Inc. (San Diego, CA, USA).

Additionally, we purchased affinity-purified goat polyclonal thyroxine 5-deiodinase (DIO2) antibody with peptide sequence EVKKHQNQEDRC from the internal region of the protein sequence as the immunogen, affinity-purified antibody to rabbit polyclonal TSH receptor with synthetic peptide directed towards the C terminal region of human TSH receptor as the immunogen, and affinity-purified mouse monoclonal thyroid-stimulating hormone that used TSH from human pituitary gland as the immunogen from Sigma Aldrich (St. Louis, MO, USA).

### 4.2. Proteins and Chemicals

In this investigation, we followed the methods of chemical binding to proteins from our previous publication, in which we ascertained elevated levels of antibodies against xenobiotics in a subgroup of healthy subjects [19]. In that study, we identified antibodies to the same chemical haptens in about 20% of tested individuals. For this study, we used the same binding methods.

We purchased the HSA, BSA, hemoglobin, formaldehyde, tolylene-2.4-diisocyanate, trimellitic anhydride, p-amino benzoic acid, bisphenol-A (BPA), tetrabromobisphenol-A (TBBPA), isopropyl benzoic acid, cyanoethyl benzoic acid, propyl 4-hydroxybenzoic acid, permethrin, mercury chloride, nickel sulfate, cobalt acetate, cadmium chloride, lead acetate, and arsenic oxide from Sigma Aldrich (St. Louis, MO, USA).

### 4.3. Preparation of Chemicals Bound to HSA

Because HSA is found in most human body tissues, and copious amounts exist in serum, we selected it as our model protein. HSA has a well-defined monomeric and three-dimensional structure. Additionally, in vivo studies by Chipinda et al. [25] reported that HSA is a target of hapten binding. We prepared formaldehyde-human serum albumin (Formal-HSA) using the method published by Plan et al. [26]. We used a similar method as published by Pezzini et al. [27]. The preparation of trimellitic + phthalic anhydride-human serum albumin (TPA-HSA) was developed by the method published by Plan et al. [26] to filter the mixture and keep it at −20 °C. The preparation of 2,4-dinitrophenol ring-HSA (Din-HSA), BPA-human serum albumin (BPA-HSA), tetrabromobisphenol-A-human serum albumin (T-BPA-HSA), tetrachlorothylene-human serum albumin (T-ethyl-HSA), parabens-human serum albumin (Para-HSA), mercury-human serum albumin (Merc-HSA), and heavy metals-human serum albumin (Hvy mtls-HSA) conjugate was achieved through methods described by Vojdani et al. [19]. These methods have confirmed conjugation of haptenic chemicals by gel electrophoresis and spectrographic analyses.

### 4.4. Demonstration of Anti-Thyroid Antibody Binding to Various Chemicals Bound to HSA Using ELISA

HSA or chemical bound to HSA at a concentration of 1.0 mg/mL^−1^ was diluted 1:100 in 0.1 M phosphate buffer saline (PBS) with a pH of 7.4. We added 100 μL of the chemical bound to HSA to different wells of microtiter ELISA plates. The plates coated with antigens were incubated overnight and then washed with 200 μL of PBS containing a pH of 7.4. Non-specific binding of antibodies was prevented by adding 2% BSA in PBS and then incubating overnight at 4 °C. The plates were washed again, and then monoclonal and polyclonal antibodies were added to the wells at an optimal dilution of 1:300 in 0.1 M PBS Tween containing 2% BSA and incubated for 1 h at room temperature. The plates were washed, and then secondary antibodies or goat anti-rabbit or anti-mouse IgG F(ab’)2 fragments (KPI, Gaithersburg, MS, USA) were added to each well at an optimal dilution of 1:200–1:1000 in 2% BSA-TBS were added to each well. The plates then were incubated for an additional 1 h at room temperature and washed with PBS-Tween buffer four times. The enzymatic response was started by adding 100 μL of substrate in diethanolamine buffer 1 mg/mL^−1^ at a pH 9.8. Using a microtiter reader, the optical densities (ODs) were read at 405 nm. For demonstration of non-specific binding of antibody to the plate coated with antigens, we used several control wells that were coated with HSA, BSA, or hemoglobulin by itself, followed by all other reagents. We measured one antibody to each individual thyroid axis component. Mouse monoclonal antibodies to TBG, thyroglobulin, T4, T3, thyroid peroxidase (TPO), goat polyclonal antibodies to DIO2, rabbit polyclonal antibodies to TSH receptor, and mouse monoclonal antibodies to TSH were used against 11 chemicals bound to HSA adducts and the enzyme protein disulfide isomerase (PDI) was used in ELISA analyses. The chemicals were all bound to HSA and included aflatoxin bound to human serum albumin (Afla-HSA), formaldehyde bound to human serum albumin (Formal-HSA), isocyanates bound to human serum albumin (Iso-HSA), trimellitic + phthalic anhydride bound to human serum albumin (TPA-HSA), 2,4-dinitrophenol bound to human serum albumin (2,4-Din-HSA), bisphenol-A bound to human serum albumin (BPA-HSA), tetrabromobisphenol-A bound to human serum albumin (T-BPA-HSA), tetrachlorothylene bound to human serum albumin (T-ethyl-HSA), mercury bound to human serum albumin (Merc-HSA), parabens bound to human serum albumin (Para-HSA), heavy metals bound to human serum albumin (Hvy mtls-HSA).

### 4.5. Binding of Serially Diluted Anti-Thyroid Antibodies with Different Haptenic Chemicals Bound to HSA

For demonstrating the specificity of anti-thyroid antibodies binding to different chemicals bound to HSA, 10 different strips of microtiter plate coated with fixed concentrations of chemicals–HSA were exposed to serially diluted different thyroid antibodies. The final concentration for each of the anti-thyroid-stimulating hormone receptor, anti-thyroxine deiodinase, anti-triiodothyronine, anti-thyroglobulin, and thyroid peroxidase antibodies ranged from 10 μ/mL to as low as 0.075 μ/mL. After proper incubation washing and addition of secondary antibody, plus completion of all other steps, the ODs were recorded at 405 nm.

### 4.6. Inhibition of Anti-Thyroid Antibodies Binding to Different Haptenic Chemicals Bound to HSA-Coated Plates with the Same Haptenic Chemicals in Liquid Phase

Two different rows of microtiter plates coated with different haptenic chemicals, 50 μL serially diluted with the same haptenic chemical concentration of 0, 2, 4, 8, 16, 32, 64, and 128 μg/mL, were added to different wells of microtiter plate and immediately the same concentration of different thyroid antibodies was added to these series of 8-well strips of microtiter plates. After shaking, incubation, and washing, secondary antibodies were added and incubated again. Following repeated washing and the addition of substrate, the color development was measured and ODs were recorded at 405 nm. These analyses were also conducted with the chemicals that were not complexed with HSA, as well as with HSA alone.

## 5. Conclusions

We propose that chemicals can bind to proteins in serum and lead to the development of new epitopes. These epitopes can lead to cross-reactive interactions with various target sites of the thyroid axis and the formation of antibodies that potentially generate immunological reactivity with various target sites of the thyroid axis, and which may promote autoimmune thyroid reactivity. The results of our study provide new insights into how hapten–protein adducts can develop into neoantigens that may lead to immunological interactions with various target proteins of the thyroid axis due to cross-reactivity and potentially cause thyroid metabolism disruption. Further research evolving into animal and human models would be necessary to investigate the exact clinical role of these interactions.

## Figures and Tables

**Figure 1 ijms-21-07324-f001:**
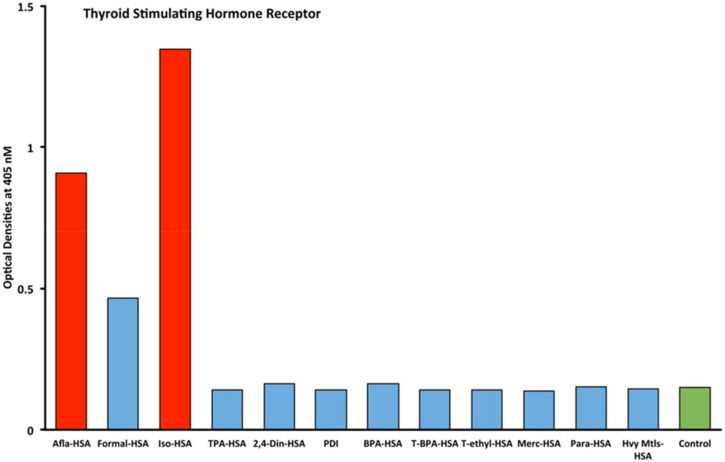
Reaction of thyroid-stimulating hormone receptor polyclonal antibodies with different chemicals bound to human serum albumin. The antigens in red represent optical density (OD) levels that are above the control and also 1 standard deviation above the mean of all 84 single measurements of chemicals bound to human serum albumin (HSA). The antigens in blue are less than 1 standard deviation above the mean. The control is in green. Afla-HSA = aflatoxin-HSA; Formal-HSA = formaldehyde-HSA; Iso-HSA = isocyanates-HSA; TPA-HSA = trimellitic + phthalic anhydride-HSA; 2,4-Din-HSA = 2,4-dinitrophenol-HSA; PDI = protein disulfide isomerase; BPA-HSA = bisphenol-A-HSA; T-BPA-HSA = tetrabromobisphenol-A-HSA; T-ethyl-HSA = tetrachloroethylene-HSA; Merc-HSA = mercury-HSA; Para-HSA = parabens-HSA; Hvy Mtls-HSA = heavy metal composite-HSA.

**Figure 2 ijms-21-07324-f002:**
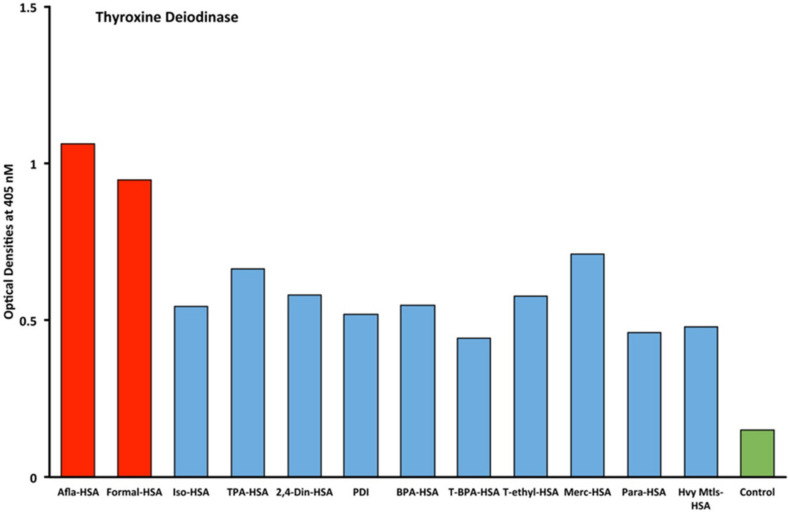
Reaction of thyroxine deiodinase polyclonal antibodies with different chemicals bound to human serum albumin. The antigens in red represent OD levels that are above the control and also 1 standard deviation above the mean of all 84 single measurements of chemicals bound to HSA. The antigens in blue are less than 1 standard deviation above the mean. The control is in green. Afla-HSA = aflatoxin-HSA; Formal-HSA = formaldehyde-HSA; Iso-HSA = isocyanates-HSA; TPA-HSA = trimellitic + phthalic anhydride-HSA; 2,4-Din-HSA = 2,4-dinitrophenol-HSA; PDI = protein disulfide isomerase; BPA-HSA = bisphenol-A-HSA; T-BPA-HSA = tetrabromobisphenol-A-HSA; T-ethyl-HSA = tetrachloroethylene-HSA; Merc-HSA = mercury-HSA; Para-HSA = parabens-HSA; Hvy Mtls-HSA = heavy metal composite-HSA.

**Figure 3 ijms-21-07324-f003:**
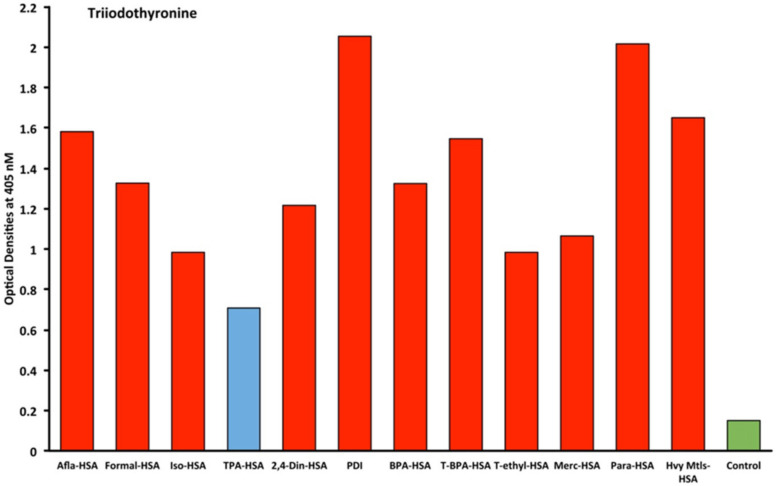
Reaction of monoclonal triiodothyronine antibodies with different chemicals bound to human serum albumin. The antigens in red represent OD levels that are above the control and also 1 standard deviation above the mean of all 84 single measurements of chemicals bound to HSA. The antigens in blue are less than 1 standard deviation above the mean. The control is in green. Afla-HSA = aflatoxin-HSA; Formal-HSA = formaldehyde-HSA; Iso-HSA = isocyanates-HSA; TPA-HSA = trimellitic + phthalic anhydride-HSA; 2,4-Din-HSA = 2,4-dinitrophenol-HSA; PDI = protein disulfide isomerase; BPA-HSA = bisphenol-A-HSA; T-BPA-HSA = tetrabromobisphenol-A-HSA; T-ethyl-HSA = tetrachloroethylene-HSA; Merc-HSA = mercury-HSA; Para-HSA = parabens-HSA; Hvy Mtls-HSA = heavy metal composite-HSA.

**Figure 4 ijms-21-07324-f004:**
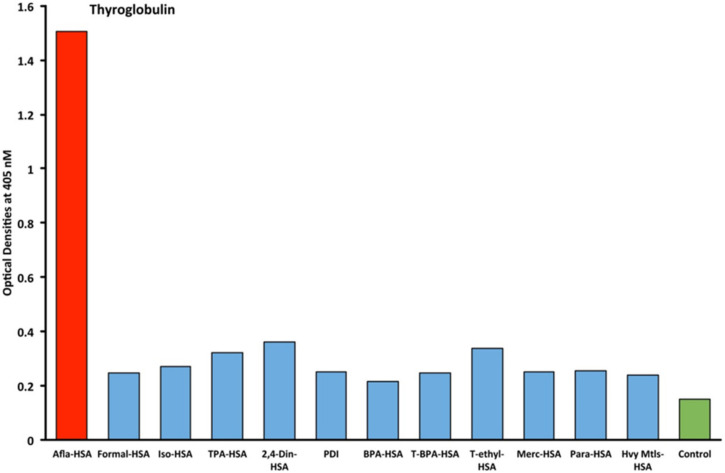
Reaction of thyroglobulin monoclonal antibodies with different chemicals bound to human serum albumin. The antigens in red represent OD levels that are above the control and also 1 standard deviation above the mean of all 84 single measurements of chemicals bound to HSA. The antigens in blue are less than 1 standard deviation above the mean. The control is in green. Afla-HSA = aflatoxin-HSA; Formal-HSA = formaldehyde-HSA; Iso-HSA = isocyanates-HSA; TPA-HSA = trimellitic + phthalic anhydride-HSA; 2,4-Din-HSA = 2,4-dinitrophenol-HSA; PDI = protein disulfide isomerase; BPA-HSA = bisphenol-A-HSA; T-BPA-HSA = tetrabromobisphenol-A-HSA; T-ethyl-HSA = tetrachloroethylene-HSA; Merc-HSA = mercury-HSA; Para-HSA = parabens-HSA; Hvy Mtls-HSA = heavy metal composite-HSA.

**Figure 5 ijms-21-07324-f005:**
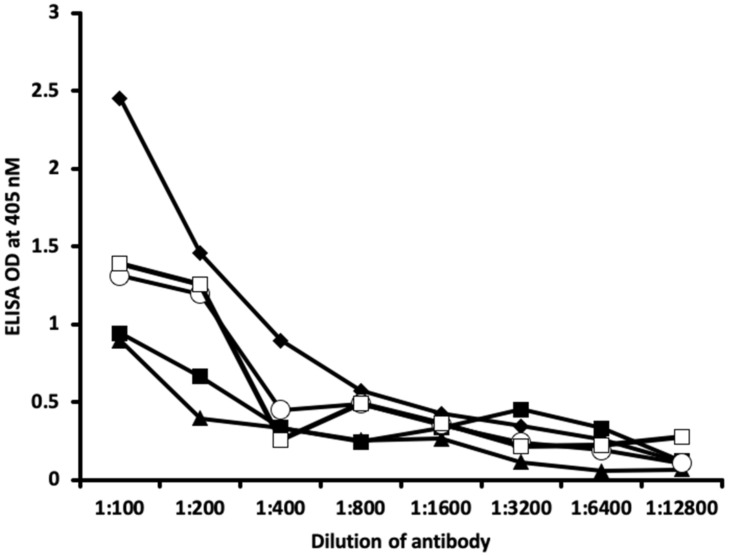
Reaction of different dilutions of thyroid antibodies with the same concentration of haptenic chemical-coated plates. Note that in proportion to the antibody dilution, a significant decline in the reaction of the antibodies to the haptenic chemicals was observed. ♦ = thyroid-stimulating hormone receptor (TSH-R) antibody to isocyanate-HSA; ■ = TSH-R antibody to aflatoxin-HSA; ▲ = TSH-R antibody to formaldehyde-HSA; ○ = thyroxine deiodinase antibody to aflatoxin-HSA; □ = thyroxine deiodinase antibody to formaldehyde-HSA.

**Figure 6 ijms-21-07324-f006:**
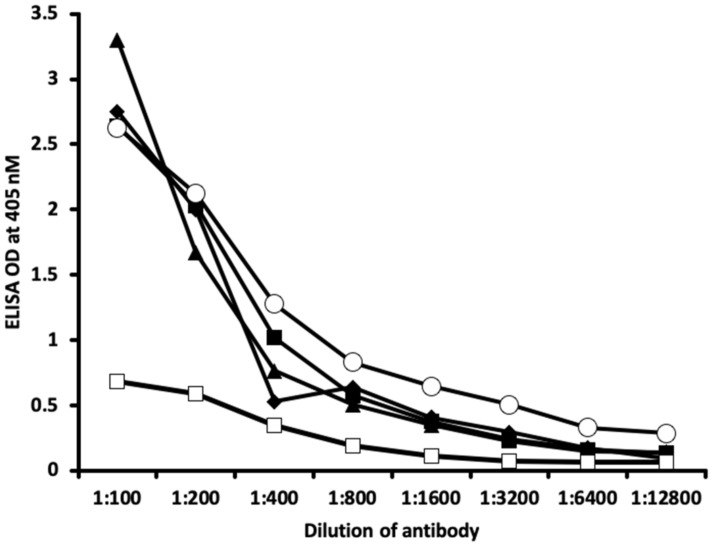
Reaction of different dilutions of thyroid antibodies with the same concentration of haptenic chemical-coated plates. Note that in proportion to the antibody dilution, a significant decline in the reaction of the antibodies to the haptenic chemicals was observed. ♦ = triiodothyronine antibody to protein disulfide isomerase (PDI); ■ = triiodothyronine antibody to parabens-HSA; ▲ = triiodothyronine antibody to aflatoxin-HSA; ○ = thyroglobulin antibody to aflatoxin-HSA; □ = thyroid peroxidase antibody to formaldehyde-HSA.

**Figure 7 ijms-21-07324-f007:**
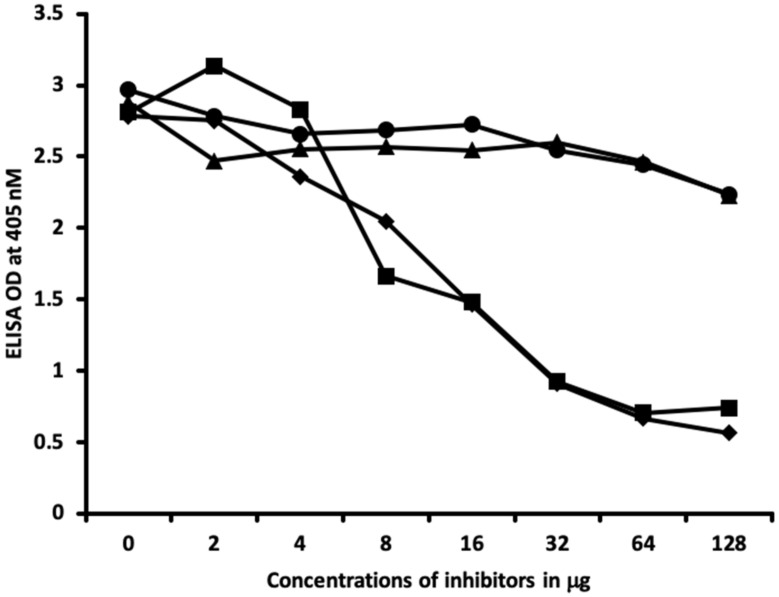
Binding of thyroid-stimulating hormone receptor antibody to isocyanate-HSA in the absence or presence of isocyanate aflatoxin, BPA-HSA, and 2,4-Din-HSA in concentrations of 0, 2, 4, 8, 16, 32, 64, and 128 μg/mL. ♦ = Isocyanate; ■ = aflatoxin-HSA; ▲ = BPA-HSA; ● = DNP-HSA.

**Figure 8 ijms-21-07324-f008:**
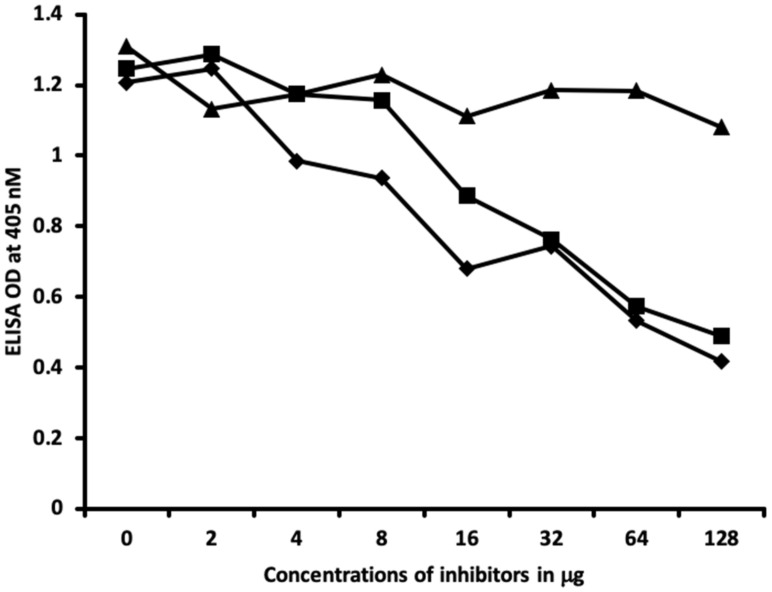
Binding of thyroxine deiodinase antibody to aflatoxin-HSA in the absence or **presence** of aflatoxin-HSA, formaldehyde-HSA, and parabens-HSA in concentrations of 0, 2, 4, 8, 16, 32, 64, and 128 μg/mL. ♦ = Aflatoxin-HSA; ■ = formaldehyde-HSA; ▲ = parabens-HSA.

**Figure 9 ijms-21-07324-f009:**
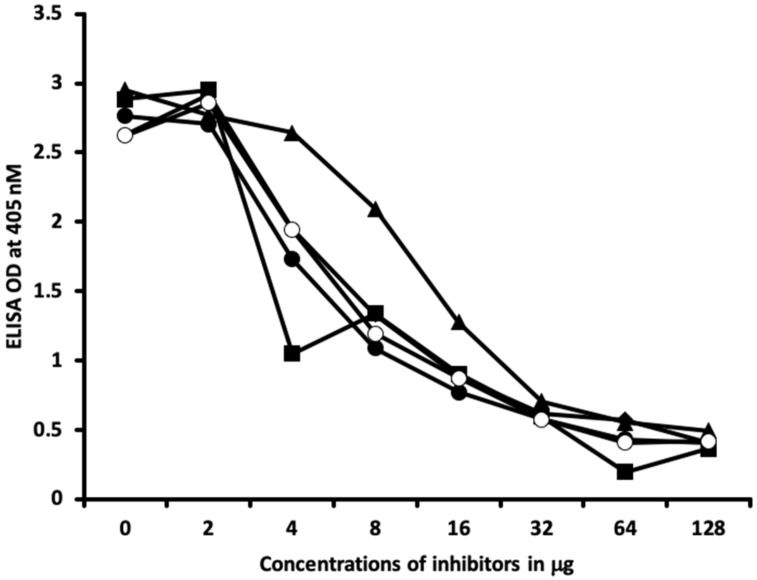
Binding of triiodothyronine antibody to parabens-HSA in the absence or presence of parabens-HSA, aflatoxin-HSA, formaldehyde-HSA, T-BPA-HSA, and TPA-HSA in concentrations of 0, 2, 4, 8, 16, 32, 64, and 128 μg/mL. ♦ = Parabens-HSA, ○ = aflatoxin-HSA; ■ = formaldehyde-HSA; ▲ = T-BPA-HSA; ● = TPA-HSA.

**Figure 10 ijms-21-07324-f010:**
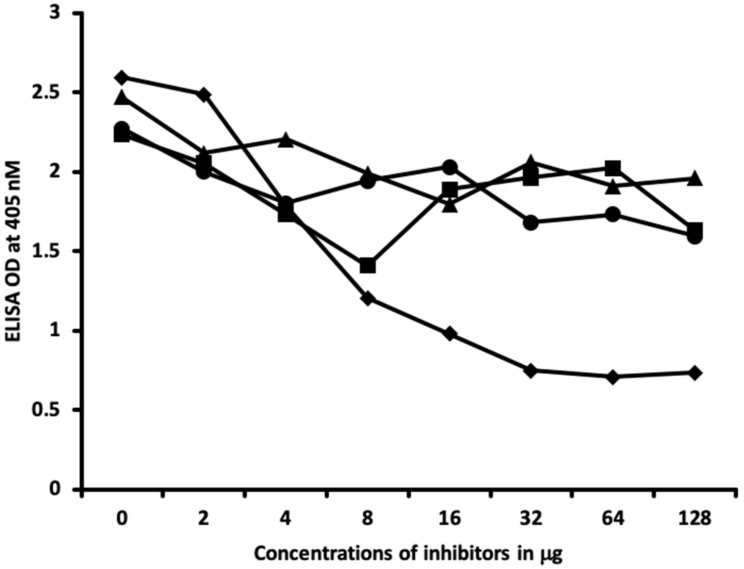
Binding of thyroglobulin antibody to aflatoxin-HSA in the absence or presence of aflatoxin-HSA, formaldehyde-HSA, BPA-HSA, and parabens-HSA in concentrations of 0, 2, 4, 8, 16, 32, 64, and 128 μg/mL. ♦ = Aflatoxin-HSA; ■ = formaldehyde-HSA; ▲ = BPA-HSA; ● = parabens-HSA.

**Figure 11 ijms-21-07324-f011:**
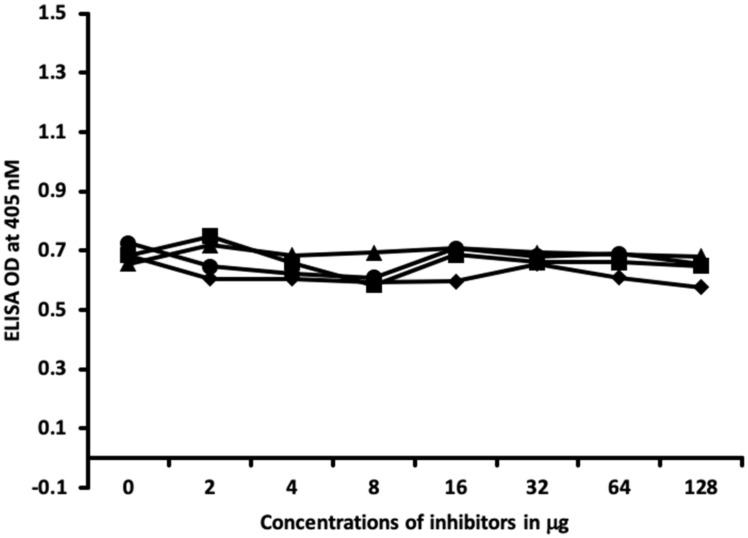
Binding of thyroid peroxidase antibody to formaldehyde-HSA in the absence or presence of formaldehyde-HSA, isocyanate-HSA, BPA-HSA, and mercury-HSA in concentrations of 0, 2, 4, 8, 16, 32, 64, and 128 μg/mL. ♦ = Formaldehyde-HSA; ■ = isocyanate-HSA; ▲ = BPA-HSA; ● = mercury-HSA.

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
