# Peer review of "Cross-Reactivity between Chemical Antibodies Formed to Serum Proteins and Thyroid Axis Target Sites"

_ijms, 2020, doi:10.3390/ijms21197324_

Round 1
Reviewer 1 Report
The authors describe experiments to modify the immunogenicity of human serum albumin using a 12 different chemical adducts. They then tested the antigenicity against 4 commercial monoclonal antibodies directed against TBG, T4 T3 TPO and and affinity purified polyclonal antibodies against DIO2, TSRH. The formation of immune-complexes was detected by ELISA by allowing the albumin adducts to adhere to microtitre plate, antibodies added and then complexes detected using labelled secondary detection antibodies (details not specified). The ELISA experiments were controlled using native human albumin, BSA and haemoglobin. The affinity and specificity of the complexes were investigated by serial dilution and competition with unbound antigen.
Using the HSA adducts to coat the ELIAS plates caused more secondary antibody to be capture by the ELISA plate than HSA alone with the TSHR DIO2 polycloncal antibodies and the T3 and Tglob monoclonals.
The ELISA signal was dependent on the dose of antibody used, showing that the signal was dependent on the presence of antibody. The signal could also be ablated by adding the original antigen used to coat the plate suggesting specific binding, although it is not quite clear as to what discp.
The authors conclude that chemical adducts can change the antigenicity of HSA, and speculate that this is due to protein misfolding generating secondary epitopes - rather than the adduct per se However the authors also conclude that the anti T3 results may be due to structural similarity between and adduct and T3 .
The observation that chemical modification of albumin alters its immunogenicity is not surprising. I think the key question to ask is whether this is an effect of the adduct itself or a modifying effect of the adduct on HAS creating a ‘neo-antigen. It is not clear to me as to whether the authors have addressed this. An obvious experiment would be to compare the native adduct and the HSA complexed adduct for the ability to ‘inhibit’ the binding of antibody to the plate. It would be necessary to demonstrate an effect of the HSA adduct over and above that of the adduct alone to invoke a neoantigan.
I’m presuming that the ‘inhibition’ experiments were done with the HAS-adduct, not the chemical alone – although this is not clear
136 – “same haptenic chemicals in liquid phase “ is this the HSA adduct or just the ‘chemical’
line 248 isocyanate –HSA,
line 259 isocyante – is this HSA isocyanate or just isocyanate?)
Showing that the chemical alone could not ‘inhibit’ binding – at least for some antibodies would add great weight to this argument.
I also not sure about the use of the term ‘molecular mimicry in this context. My understanding is that some proteins expressed by pathogens have an evolutionary advantage by being similar to host proteins. I don’t see that a chemical adduct quite fits the bill, although clearly an adduct could provoke an immune response that could cause automimmunity.
It is also difficult to estimate the affinity of the neo-antigen from the ‘inhibition curves’ – most antibodies are likely to be ‘heterophiles’ and bind other epitopes albeit weakly, whilst I don’t think it is necessary to provide formal estimates of affinity – some discussion is required. Given ugml concentrations of inhibitors required to displace these are likely to be quite weak interactions.
In general I found the paper quite difficult to read and the authors predisposed to over-interpret the data. I think the paper would benefit from statement of a clear hypothesis, without conjecture, which should be left to the discussion.
Reviewer 2 Report
The authors are studying the wrong binding proteins. For T4 and T3 should study TBG which is the high affinity binding protein. Also effects of other agents on T4 and T3 concentrations should be studied employing both immunoassay and mass spectrometric methods.
Reviewer 3 Report
I found this to be an interesting well-conceived and well written paper outlining how certain chemicals can interact with human albumin to make it immunogenic and thus potentially cross react with autoantibodies that would otherwise be directed against thyroid epitopes.
In addition to showing the effects these new antibodies may have on thyroid function itself, this paper also alludes to the effects that they may have on treatment of thyroid disease.
What I am less clear about from the data in the paper would be how these in-vitro findings translate into an in-vivo situation. The authors have previous work showing that similar antibodies can be detected in some healthy subjects but can we be certain that exposure of human albumin to these chemicals in-vivo will result in antibody formation and will those antibodies behave in the same manner in the body as they do on an ELISA plate?
What might be useful too would be for the authors to speculate on the two following areas that seem to me to flow naturally from their findings:
(i) what would be the effect on making a diagnosis of Grave’s Disease in which one would expect to see primary antibodies to TSH receptor if these cross-reacting antibodies are present? Is it too simplistic for me to ask if there is any possibility of a false positive result? I know that the TFTs themselves would provide compelling evidence of the need to treat with carbimazole or propylthiouracil but the antibody status can affect management.
(ii) What would be the effect of the development of these antibodies on the assays that detect thyroid related antibodies? Once again, do we know if the presence of cross-reacting antibodies will give false positive results? If so, for example, this could have profound effects on the decision to treat subclinical hypothyroidism with thyroxine when, as clinicians, we are looking for a positive thyroid peroxidase antibody to give us a bit more confidence that starting someone on thyroxine replacement is the best thing to do!
One also wonders what other antibodies might be cross-reacting with other endocrine organ systems.... for example, might they cross react with antibodies to parathyroid tissue, might the diagnosis or Addison’s or Cushing’s Diseases be similarly affected?
I would also suggest the following typo changes that make the sentences clearer.
Line 126 – replace the word “containing” with the word “with”
Line 421 – replace the word “These” with the word “This”.
Line 439 - replace the word “he” with the word “the”.
On a personal note, I found the deleted text that was still visible, the different colours and underlining in the text (presumably denoting text that had been added) quite distracting!
Author Response
Dear Reviewer #3,
Thank you for taking the time to review and comment on our manuscript.
We have downloaded the latest version of the manuscript, but there may have been some discrepancies with the lines of the manuscript you commented on and the lines on our version of the manuscript. We also feel the editor may have made the corrections you requested already.
We have made the following changes as requested by your comments to the best of our abilities.
Comment # 1: Line 126 – replace the word “containing” with the word “with”
Response #1 – Line 126 does not state the word “containing” in the manuscript. Only lines 144 and 147 have the word “containing” and on those lines, we switched the word to “with” instead of “containing.” These changes have improved the readability of the manuscript.
Comment #2: Line 421 – replace the word “These” with the word “This.”
Response #2. Line 421 does not have the word “These,” so we searched the entire document for proper conversion of “These” and “This,” and on line 476, we changed “These finding” to “These findings.” Thank you for bringing that to our attention.
Comment #3: Line 439 - replace the word “he” with the word “the”
Response #3: There is no line 439 in our version of the manuscript so we searched the entire document for the improper use of “he” for “the” and we were not able to find any errors at this time. It is possible that this has been corrected already.
Comment #4: What I am less clear about from the data in the paper would be how these in-vitro findings translate into an in-vivo situation. The authors have previous work showing that similar antibodies can be detected in some healthy subjects but can we be certain that exposure of human albumin to these chemicals in-vivo will result in antibody formation and will those antibodies behave in the same manner in the body as they do on an ELISA plate?
Response #4: This is an important question that you have asked. Unfortunately, we do not have the study design or data to provide any commentary regarding "in-vivo" questions. We hope the publication of our manuscript will allow other researchers to consider exploring the "in-vivo" applications. Our study was limited to the initial laboratory analysis to consider this possibility. We did state in our conclusion, “further research evolving into animal and human models would be necessary to investigate the exact clinical role of these interactions.”
Comment #5: What might be useful too would be for the authors to speculate on the two following areas that seem to me to flow naturally from their findings:
(i) What would be the effect on making a diagnosis of Grave’s Disease in which one would expect to see primary antibodies to TSH receptor if these cross-reacting antibodies are present? Is it too simplistic for me to ask if there is any possibility of a false positive result? I know that the TFTs themselves would provide compelling evidence of the need to treat with carbimazole or propylthiouracil but the antibody status can affect management.
Response#5: We can only speculate that cross-reactivity has the potential for false positives; however, we do not have the data to support this speculation, and we would need to consider this hypothesis for a separate study design. We did state in our conclusion, “further research evolving into animal and human models would be necessary to investigate the exact clinical role of these interactions.”
Comment #6: (ii) What would be the effect of the development of these antibodies on the assays that detect thyroid-related antibodies? Once again, do we know if the presence of cross-reacting antibodies will give false-positive results? If so, for example, this could have profound effects on the decision to treat subclinical hypothyroidism with thyroxine when, as clinicians, we are looking for a positive thyroid peroxidase antibody to give us a bit more confidence that starting someone on thyroxine replacement is the best thing to do!
Response #6. This is another essential question that will need further investigation. Our laboratory study can only provide the first step to answer your question. The goal of our study design was limited to only identify the cross-reactive interactions as a first step. Further research will be needed to determine the actual clinical implications of these cross-reactive interactions. We did state in our conclusion, “further research evolving into animal and human models would be necessary to investigate the exact clinical role of these interactions.”
Thank you for taking the time to review and comment on our manuscript and for understanding the importance of these cross-reactive patterns. We believe the data in our study will start the first step to answer the important clinical questions that you have raised. We hope you can provide us with the support we need to publish our data and to start the process.
Thank you

Round 2
Reviewer 1 Report
The authors still state "When chemicals bind to proteins, they induce a conformational change in the macromolecule that has the potential to misfold in such a way that makes it becomes similar to the various target sites"
Whilst I don't doubt that binding of some chemicals to proteins cause conformational change - this is not the case for all chemical adducts. There is no data provided in the paper to demonstrate conformation change in the proteins used. A neoantigen can be generated without conformation change
Author Response
Dear Reviewer #1,
Comment #1 The authors still state "When chemicals bind to proteins, they induce a conformational change in the macromolecule that has the potential to misfold in such a way that makes it becomes similar to the various target sites"
Whilst I don't doubt that binding of some chemicals to proteins cause conformational change - this is not the case for all chemical adducts. There is no data provided in the paper to demonstrate conformation change in the proteins used. A neoantigen can be generated without conformation change
Response #2: We agree with your point and we made the following change to the manuscript to be more accurate:
Changes on Abstract:
"In some instances, when chemicals bind to proteins, they have the potential to induce a conformational change in the macromolecule that may misfold in such a way that makes it similar to the various target sites or act as a neoantigen without conformational change."
Changes in Conclusion:
We removed the words, "...leading to protein misfolding and the development of new epitopes." and change the sentence to: "We propose that chemicals can bind to proteins in serum and lead to the development of new epitopes."
Thank you for helping us make this important distinction and improving our manuscript.
Reviewer 2 Report
-
Author Response
Dear Reviewer 3,
We have improved the background, methods, and results section of our manuscript in addition to making the necessary minor grammar and spelling changes that were recommended by all of the reviewers.
Thank you for your comments and for taking the time to review our manuscript.